# Diverse Image Captioning with Context-Object Split Latent Spaces

**Shweta Mahajan**     **Stefan Roth**
Dept. of Computer Science, TU Darmstadt
{mahajan@aiphes, stefan.roth@visinf}.tu-darmstadt.de

## Abstract

Diverse image captioning models aim to learn one-to-many mappings that are innate to cross-domain datasets, such as of images and texts. Current methods for this task are based on generative latent variable models, *e.g.* VAEs with structured latent spaces. Yet, the amount of multimodality captured by prior work is limited to that of the paired training data – the true diversity of the underlying generative process is not fully captured. To address this limitation, we leverage the contextual descriptions in the dataset that explain similar contexts in different visual scenes. To this end, we introduce a novel factorization of the latent space, termed *context-object split*, to model diversity in contextual descriptions across images and texts within the dataset. Our framework[1] not only enables diverse captioning through context-based pseudo supervision, but extends this to images with novel objects and without paired captions in the training data. We evaluate our *COS-CVAE* approach on the standard COCO dataset and on the held-out COCO dataset consisting of images with novel objects, showing significant gains in accuracy and diversity.

## 1   Introduction

Modeling cross-domain relations such as that of images and texts finds application in many real-world tasks such as image captioning [5, 9, 20, 23, 26, 31, 33, 48]. Many-to-many relationships are innate to such cross-domain tasks, where a data point in one domain can have multiple possible correspondences in the other domain and vice-versa. In particular for image captioning, given an image there are many likely sentences that can describe the underlying semantics of the image. It is thus desirable to model this multimodal conditional distribution of captions (texts) given an image to generate plausible, varied captions.

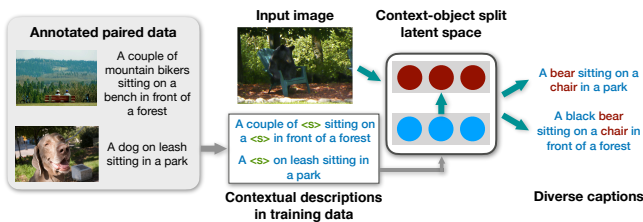

Figure 1: Context-object split latent space of our COS-CVAE to exploit similarities in the contextual annotations for diverse captioning.

Recent work has, therefore, explored generating multiple captions conditioned on an image to capture these diverse relationships in a (conditional) variational framework [6, 14, 32, 42]. These variational frameworks encode the conditional distribution of texts given the image in a low-dimensional latent space, allowing for efficient sampling. However, the diversity captured by current state-of-the-art latent variable models, *e.g.* [6, 42], is limited to the paired annotations provided for an image. While the recently proposed LNFMM approach [32] allows for additional unpaired data of images or texts to be incorporated, the amount of diversity for a given image is again limited to the paired annotated data.

Since standard learning objectives reward only the generated captions that belong to ground-truth annotations, the underlying multimodality of the conditional image and text distributions in the latent space can be better captured with access to more annotated paired data. To this end, the initial work of Kalyan et al. [21] improves multimodality of the input-output mappings with additional supervision from annotations of 'nearby' images. However, Kalyan et al. model one-to-many relationships in the posterior distribution, where the sampling time grows exponentially in the sequence length. Thus beam search is required for sampling diverse captions, which makes it computationally inefficient [5].

In this work, we propose a novel factorized latent variable model, termed *context-object split conditional variational autoencoder (COS-CVAE)*, to encode object and contextual information for image-text pairs in a factorized latent space (Fig. 1). We make the following contributions: *(i)* Our COS-CVAE framework exploits contextual similarities between the images and captions within the dataset in a pseudo-supervised setup. Specifically, the COS factorization in addition to annotated paired data leverages diverse contextual descriptions from captions of images that share similar contextual information, thus encoding the differences in human captions that can be attributed to the many ways of describing the contexts. *(ii)* This additionally allows extending COS-CVAE to previously unseen (novel) objects in images. To the best of our knowledge, this is the first (variational) framework that allows to describe images with novel objects in a setup for diverse caption generation. *(iii)* We show the benefits of our approach on the COCO dataset [29], with a significant boost in accuracy and diversity. Moreover, varied samples can be efficiently generated in parallel.

## 2 Related Work

**Image captioning.**  Classical approaches for image captioning are framed as sequence modeling task, where image representations from convolutional neural networks (CNNs) are input to recurrent neural networks (RNNs) to generate a single caption for an image [15, 19, 23, 30, 33, 41, 46]. Instead of RNNs, recent work uses convolutional networks [5] or transformer-based networks [11] for caption generation. Anderson et al. [1] have proposed an attention-based architecture, which attends to objects for generating captions. Lu et al. [31] generate caption templates with slots explicitly tied to image regions, which are filled with objects from object detectors [35]. These methods optimize for single accurate captions for an image and, therefore, cannot model one-to-many relationships.

**Diverse image captioning.**  Recent work on image captioning has focused on generating multiple captions to describe an image. Vijayakumar et al. [40] generate diverse captions based on a word-to-word Hamming distance in the high-dimensional output space of captions. Kalyan et al. [22] improve the accuracy of [40] by optimizing for a set of sequences. Cornia et al. [10] predict saliency maps for different regions in the images for diverse captioning. Wang and Chan [43] encourage diversity using a determinantal point process. To mitigate the limitations of sampling in high-dimensional spaces, recent work leverages generative models such as generative adversarial networks (GANs) [36] or Variational Autoencoders (VAEs). Since GANs for diverse caption generation have limited accuracy, recent works [6, 32, 44] have used VAEs to learn latent representations conditioned on images to sample diverse captions. Wang et al. [44] learn latent spaces conditioned on objects whereas Chen et al. [8] leverage syntactic or lexical domain knowledge. Aneja et al. [6] enforce sequential Gaussian priors and Mahajan et al. [32] learn domain-specific variations in a latent space based on normalizing flows. In all cases, the amount of diversity that can be modeled is limited to the availability of paired training data. Here, in contrast, we extend the task of diverse image captioning to exploit information from the annotations in the dataset through context-based pseudo supervision.

**Novel object captioning.**  Recent works [3, 12, 17, 27, 31, 39, 45, 47] have focused on generating a single accurate caption containing novel objects, for which no paired training data is provided. Many proposed approaches [27, 31, 47] explicitly use object detectors at test time to generate captions with novel objects. Lu et al. [31] generate slotted captions, where the slots are filled with novel objects using object detectors. Li et al. [27] extend the copying mechanism of Yao et al. [47] with pointer networks to point where to copy the novel objects in the caption and Cao et al. [7] use meta learning for improved accuracy. In contrast, our framework does not require class labels of novel objects at test time. Constrained beam search [2] based on beam search enforces the generated sequence to contain certain words. Anderson et al. [3] use constrained beam search where objects detected from images serve as constraints to be included in the generated captions for novel objects, which are then used for training to generate a single caption. In this work, we extend novel object captioning to the task of diverse captioning with our COS-CVAE through context-based pseudo supervision.

Table 1: Context-based pseudo supervision from captions of the COCO dataset.

| Image | Human annotations | Retrieved contexts | Generated pseudo captions |
|---|---|---|---|
| 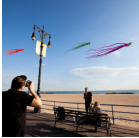 | • a man taking a picture of people on a bench near a beach<br>• there are many kites that are being flown on the beach<br>• a man is taking photos of kites in the air<br>• a man is taking a picture of people flying kites<br>• a person taking a photo of a kite in the sky | • a \<s\> holding on to a \<s\> while \<s\> sail in the background<br>• a \<s\> flying a \<s\> in the blue cloud filled sky<br>• large \<s\> are being flown in the air<br>• several octopus \<s\> are being flown in a blue sky<br>• two \<s\> flying in the sky next to a smoke stack | • a person holding on to a bench while kites sail in the background<br>• a person flying a kite in the blue cloud filled sky<br>• large kite are being flown in the air<br>• several octopus kites are being flown in a blue sky<br>• two kites flying in the sky next to a smoke stack |

## 3 COS-CVAE Approach

Consider the example in Tab. 1. As shown, the COCO dataset [29] provides provides five human annotations for the image (1st and 2nd column). However, there are multiple possible contextual descriptions in the dataset that can describe the image similarly well. For example, the training dataset contains a caption "a *woman* holding onto a *kite* while *sailboats* sail in the background" for an image (distinct from that in Tab. 1). The contextual description for this caption (3rd column) – where the objects in the caption are replaced with a placeholder – can describe the context of the image in Tab. 1, combining it with the concepts "person", "bench", and "kite". In this work, we aim to utilize these contexts in the latent space to capture one-to-many relationships for image captioning. Our novel COS-CVAE framework is able to better capture diversity through generated context-based pseudo supervision (4th column).

**Preliminaries.** We first split each ground-truth caption from the paired data as $x = \{x^m, x^o\}$, where $x^o$ represents the objects in the visual scene that are described in the caption while $x^m$ is the contextual description in the caption. Note that contextual description here refers to the scene information of the image excluding the objects or visible concepts. This can include the scene background, (spatial) relationships between the objects, or attributes descriptive of the objects in the image (Tab. 1, 3rd column). To provide context-based pseudo supervision, in a first step we use the method of [16] to learn a joint embedding of the paired image-(con)text data $\{x^m, I\}$ on the training set. Importantly, we encode these similarities using contextual descriptions $x^m$ instead of the caption $x$ itself. For an image $I'$ in the training set, with or without human annotated captions, this *contextual embedding* allows us to then find the nearest neighboring contextual descriptions $y^m \in N(I')$. Equipped with these additional descriptions, we can eventually overcome the limited expressiveness and variability of the annotated paired data in the dataset.

**Problem formulation.** To train a conditionally generative model for image captioning, one typically aims to maximize the conditional likelihood[2] of the conditional distribution $p_\phi(x \mid I)$, parameterized by $\phi$, for the caption sequence $x = (x_1, \dots, x_T)$ of length $T$ given an image $I$. As one of our principal aims here is to ensure diverse image-text mappings, we need to capture the multimodality of this generative process. To that end, we introduce explicit sequential latent variables $z = (z_1, \dots, z_T)$ and define the latent variable model over the data $\{x, I\}$ as

$$p_\phi(x, z \mid I) = \prod_t p_\phi(x_t \mid x_{<t}, z_{\leq t}, I)\, p_\phi(z_t \mid z_{<t}, x_{<t}, I), \tag{1}$$

where we have applied the product rule to obtain the temporal factorization (*w.l.o.g.*). The formulation of the latent distribution is crucial for encoding and capturing the multimodality of the underlying distribution, hence we introduce a novel factorization of the latent space that can leverage variations in the contextual representations within the dataset. Specifically, we factorize the latent variable $z_t = [z_t^m, z_t^o]$ at each timestep $t$ to separately encode the contextual and object information, conditioned on the given image, where $z_t^m$ is encoded independently of $x^o$. This allows our framework to utilize contextual descriptions from captions explaining similar contexts but not necessarily the same objects in a visual scene. We use an attention-based decoder [4] to model the posterior $p_\phi(x_t \mid x_{<t}, z_{\leq t}, I)$. With this attention-based posterior, the latent vector $z_t$ encoding the contextual and object information at each timestep guides the attention module of the decoder [4] to attend to regions in the image when modeling the distribution of $x_t$. The benefit of our proposed factorization is two-fold: First, we can exploit different contextual descriptions describing similar contexts in images within the dataset

to improve the diversity of the generated captions. Second, this allows us to extend our model to captions for novel objects with $z_t^o$ guiding the attention module to attend to regions in the image containing novel objects given the context $z_t^m$. Next, we formalize our proposed *context-object split (COS)* factorization of the latent space in a variational framework.

## 3.1 The log-evidence lower bound

We maximize the log-likelihood under the conditional distribution $p_\phi(x \mid I)$ from Eq. (1) in a variational framework [25]. The log-evidence lower bound (ELBO) consists of the data log-likelihood term and the KL-divergence between the posterior and a conditional prior at each timestep,

$$\log p_\phi(x \mid I) \geq \mathbb{E}_{q_\theta(z|x,I)} \left[ \sum_t \log p_\phi(x_t \mid x_{<t}, z_{\leq t}, I) \right]$$
$$- \sum_t D_{\text{KL}}\big(q_\theta(z_t \mid z_{<t}, x, I) \,\|\, p_\phi(z_t \mid z_{<t}, x_{<t}, I)\big). \tag{2}$$

Next, we introduce our posterior distribution $q_\theta$ and conditional prior distribution $p_\phi$, parameterized by $\theta$ and $\phi$ respectively, for the latent variables $z_t = [z_t^m, z_t^o]$ in our variational inference framework.

**The context-object split posterior.** The posterior $q_\theta(z \mid x, I)$ in the latent space is factorized into $q_{\theta^m}$ and $q_{\theta^o}$, encoding contextual description $x^m$ and object description $x^o$ at each timestep, parameterized by $\theta^m$ and $\theta^o$ respectively, with $\theta = \{\theta^m, \theta^o\}$,

$$q_\theta(z \mid x, I) = \prod_t q_\theta(z_t \mid z_{<t}, x, I) \tag{3a}$$

$$= \prod_t q_\theta(z_t^o, z_t^m \mid z_{<t}^o, z_{<t}^m, x, I) \tag{3b}$$

$$= \prod_t q_{\theta^o}(z_t^o \mid z_{<t}^o, z_{\leq t}^m, x^o, I) \, q_{\theta^m}(z_t^m \mid z_{<t}^m, x^m, I). \tag{3c}$$

In Eq. (3c), we assume that $z_t^m$ is conditionally independent of $z_{<t}^o$ and $x^o$ given $x^m$. Similarly, $z_t^o$ is assumed to be conditionally independent of $x^m$ given $z_{\leq t}^m$ and $z_{<t}^o$. This factorization of the posterior is chosen such that we can encode the contextual information in the scene conditionally independently of the object information in the scene. Specifically, the posterior $q_{\theta^m}(z_t^m \mid z_{<t}^m, x^m, I)$ at each timestep is conditioned on the contextual description $x^m$, allowing to encode the contexts of the scene (*cf.* Tab. 1) directly in the latent space, independently of the objects described in the particular scene. The dependence of $q_{\theta^m}$ on $z_{<t}^m$ and $x^m$ implies that the posterior encodes the likely (future) contexts for a visual scene given the previously observed context. The posterior $q_{\theta^o}(z_t^o \mid z_{<t}^o, z_{\leq t}^m, x^o, I)$ at each timestep encodes the likely object information in the scene. As we will see in Sec. 3.2, this factorization of the posterior enables us to leverage contextual descriptions for an image beyond human annotations (Tab. 1, 3rd column) by encoding retrieved contextual descriptions in $q_{\theta^m}$.

**The context-object split prior.** Equipped with the *context-object split* factorization of the latent variable $z_t = [z_t^m, z_t^o]$ and the latent variable model from Eq. (1), our COS conditional prior at each timestep, $p_\phi(z_t \mid z_{<t}, x_{<t}, I)$, factorizes as

$$p_\phi(z_t \mid z_{<t}, x_{<t}, I) = p_{\phi^o}(z_t^o \mid z_{<t}^o, z_{\leq t}^m, x_{<t}, I) \, p_{\phi^m}(z_t^m \mid z_{<t}^m, x_{<t}, I). \tag{4}$$

Here, $\phi = \{\phi^m, \phi^o\}$ are the parameters of the conditional priors $p_{\phi^m}$ and $p_{\phi^o}$, respectively. Note that we assume conditional independence of $z_t^m$ from $z_{<t}^o$ given $z_{<t}^m$, *i.e.* we do not need latent object information to specify the context prior. That is, the conditional prior $p_{\phi^m}(z_t^m \mid z_{<t}^m, x_{<t}, I)$ autoregressively encodes at each timestep the likely contextual information of the observed scenes based on the words sampled at previous timesteps (and the image). Given the contextual information $z_t^m$, the conditional prior $p_{\phi^o}(z_t^o \mid z_{<t}^o, z_{\leq t}^m, x_{<t}, I)$ encodes the object information required to generate the word at the particular timestep.

Under our factorization of the posterior and conditional priors from Eqs. (3) and (4), the KL-divergence term decomposes at every timestep as

$$D_{\text{KL}}\big(q_\theta(z_t|z_{<t}, x, I) \,\|\, p_\phi(z_t|z_{<t}, x_{<t}, I)\big) = D_{\text{KL}}\big(q_{\theta^m}(z_t^m|z_{<t}^m, x^m, I) \,\|\, p_{\phi^m}(z_t^m|z_{<t}^m, x_{<t}, I)\big) \tag{5}$$
$$+ D_{\text{KL}}\big(q_{\theta^o}(z_t^o|z_{<t}^o, z_{\leq t}^m, x^o, I) \,\|\, p_{\phi^o}(z_t^o|z_{<t}^o, z_{\leq t}^m, x_{<t}, I)\big).$$

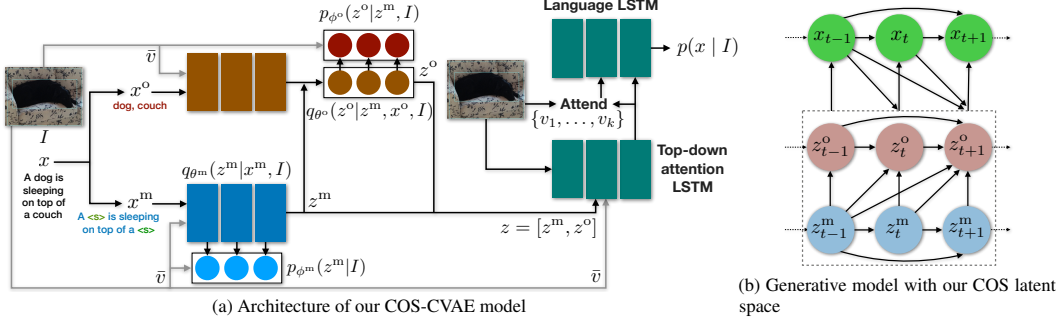

(a) Architecture of our COS-CVAE model

(b) Generative model with our COS latent space

Figure 2: Diverse image captioning with our COS-CVAE model. *(a)* Neural network architecture with posteriors and conditional priors in two sequential latent spaces for encoding objects or contexts in a visual scene. *(b)* The generative model allowing for sequential sampling in the contextual latent space of $z^{\mathrm{m}}$ and the object-based latent space $z^{\mathrm{o}}$ for generating a caption sequence.

With $z_t = [z_t^{\mathrm{m}}, z_t^{\mathrm{o}}]$ and using Eqs. (2) and (5), the final ELBO, $L(x, I)$, with our sequential context-object split decomposition of the latent space is given as

$$
\begin{aligned}
\log p_\phi(x \mid I) \geq \mathbb{E}_{q_\theta(z|x,I)} &\left[ \sum_t \log p_\phi(x_t \mid x_{<t}, z_{\leq t}, I) \right] \\
&- \sum_t D_{\mathrm{KL}}\big( q_{\theta^{\mathrm{m}}}(z_t^{\mathrm{m}} \mid z_{<t}^{\mathrm{m}}, x^{\mathrm{m}}, I) \,\big\|\, p_{\phi^{\mathrm{m}}}(z_t^{\mathrm{m}} \mid z_{<t}^{\mathrm{m}}, x_{<t}, I)) \\
&- \sum_t D_{\mathrm{KL}}\big( q_{\theta^{\mathrm{o}}}(z_t^{\mathrm{o}} \mid z_{<t}^{\mathrm{o}}, z_{\leq t}^{\mathrm{m}}, x^{\mathrm{o}}, I) \,\big\|\, p_{\phi^{\mathrm{o}}}(z_t^{\mathrm{o}} \mid z_{<t}^{\mathrm{o}}, z_{\leq t}^{\mathrm{m}}, x_{<t}, I)) = L(x, I).
\end{aligned}
\tag{6}
$$

## 3.2 Context-based pseudo supervision

Next, we show how our COS-CVAE equipped with our context-object split latent space leverages annotations of captions $N(I)$, where $N(I)$ are the "neighboring" contextual descriptions explaining similar contexts retrieved from the joint contextual embedding space (see Preliminaries, above). Consider an image $I$ with paired caption $x$ and let $y^{\mathrm{m}} \in N(I)$ be a contextual description of sequence length $T$ such that $y^{\mathrm{m}}$ contains contextual information at timesteps $t'$ and the placeholders <s> at timesteps $t''$ (see Tab. 1, 3$^{\mathrm{rd}}$ column). For supervised learning using the ELBO (Eq. 6), we require object information at timesteps $t''$. Therefore, we construct a context-based pseudo-supervision signal $\hat{y} = \{y^{\mathrm{m}}, \hat{y}^{\mathrm{o}}\}$ from the retrieved neighboring context. To obtain the object information $\hat{y}^{\mathrm{o}}$, we leverage $C(I)$, the set of objects in the image $I$ corresponding to the image regions from Faster R-CNN [35], and design the following procedure to construct $\hat{y}_{t''}^{\mathrm{o}} \in C(I)$. We encode $z_t^{\mathrm{m}}$ directly from $y^{\mathrm{m}}$ independent of the object information, facilitated by the conditional independence assumptions of COS, and leverage the conditional prior $p_{\phi^{\mathrm{o}}}$ to obtain $z_t^{\mathrm{o}}$. Note that, when training on paired training data, given $z_t = [z_t^{\mathrm{m}}, z_t^{\mathrm{o}}]$ our model learns to point to regions in the image relevant for generating the caption at each timestep. We can thus identify objects $\hat{y}_{t''}^{\mathrm{o}}$ likely to occur at timesteps $t''$ through the region attended to at that timestep, given the context $y^{\mathrm{m}}$ and latent variables $z_t$. The target caption then assumes the form $\hat{y} = \{y^{\mathrm{m}}, \hat{y}^{\mathrm{o}}\}$ (*cf.* Tab. 1, col. 4). The data term in Eq. (6) for pseudo supervision from $\hat{y}$ and image $I$ is then computed with

$$
\prod_t p_\phi(\hat{y}_t \mid \hat{y}_{<t}, z_{\leq t}, I) = \prod_{t'} p_\phi(y_{t'}^{\mathrm{m}} \mid y_{<t'}^{\mathrm{m}}, z_{\leq t'}^{\mathrm{o}}, z_{\leq t'}^{\mathrm{m}}, I) \prod_{t''} p_\phi(\hat{y}_{t''}^{\mathrm{o}} \mid y_{<t''}^{\mathrm{m}}, z_{\leq t''}^{\mathrm{o}}, z_{\leq t''}^{\mathrm{m}}, I).
\tag{7}
$$

To encode the variation in contextual descriptions given the image from the pseudo-supervision signal, we additionally enforce the posterior to match the conditional prior $p_{\phi^{\mathrm{m}}}$ by minimizing the KL divergence $D_{\mathrm{KL}}\big( q_{\theta^{\mathrm{m}}}(z_t^{\mathrm{m}} \mid z_{<t}^{\mathrm{m}}, y^{\mathrm{m}}, I) \,\big\|\, p_{\phi^{\mathrm{m}}}(z_t^{\mathrm{m}} \mid z_{<t}^{\mathrm{m}}, y_{<t}^{\mathrm{m}}, I))$. Note that the posterior $q_{\theta^{\mathrm{o}}}$ is not utilized for encoding the latent variable $z^{\mathrm{o}}$. Instead, $z^{\mathrm{o}}$ is directly obtained from the informative conditional prior $p_{\phi^{\mathrm{o}}}$ learnt on the paired training data $\{x, I\}$ and, therefore, contains object information for the image $I$. Thus, the ELBO, $\hat{L}(\hat{y}, I)$, for the augmented data point $\{\hat{y}, I\}$ is

$$
\hat{L}(\hat{y}, I) = \mathbb{E}_{q_{\theta^{\mathrm{m}}}, p_{\phi^{\mathrm{o}}}} \big[ \log p_\phi(\hat{y} \mid z, I) \big] - \sum_t D_{\mathrm{KL}}\big( q_{\theta^{\mathrm{m}}}(z_t^{\mathrm{m}} | z_{<t}^{\mathrm{m}}, y^{\mathrm{m}}, I) \,\big\|\, p_{\phi^{\mathrm{m}}}(z_t^{\mathrm{m}} | z_{<t}^{\mathrm{m}}, y_{<t}^{\mathrm{m}}, I)).
\tag{8}
$$

Taken together, our context-object split latent space can be leveraged to overcome the limited paired annotations by augmenting with neighboring captions where different images can share contextual information despite showing different objects, thereby encouraging multimodality in the latent space.

**Extension to novel objects.** Another benefit of the proposed COS factorization is that it allows to generate diverse captions for images with novel objects. Consider an image $I$ containing novel objects for which no caption annotations are available in the dataset. We first retrieve contextual descriptions $y^{\mathrm{m}} \in N(I)$ as above. To construct the supervision signal, we use $y^{\mathrm{m}}$ and $C_k(I)$, the top-$k$ objects from Faster R-CNN [35], including novel objects in the image $I$. However, the latent space of $z^{\mathrm{o}}$ learnt from the paired training data may be suboptimal (observed empirically) for attending to regions in an image with novel objects. We thus learn a weaker posterior $q_{\theta^{\mathrm{o}}}(z_t^{\mathrm{o}} \mid z_{<t}^{\mathrm{o}}, z_{\leq t}^{\mathrm{m}}, C_k(I), I)$, conditioned on $C_k(I)$ instead of $x^{\mathrm{o}}$ otherwise available from paired training data, at each timestep during training. With this formulation of the posterior and $y^{\mathrm{m}}$, we encode the representation $[z_t^{\mathrm{m}}, z_t^{\mathrm{o}}]$, which guides the attention LSTM to attend to regions in the image, including regions with novel objects, $\hat{y}^{\mathrm{o}} \in C_k(I)$, to generate the word at a certain timestep. Here, we assume that the description $y^{\mathrm{m}}$ has been observed in the paired training data and thus $z_t^{\mathrm{m}}$ can guide the attention LSTM to attend to salient regions in the image. The target caption is of the form $\hat{y} = \{y^{\mathrm{m}}, \hat{y}^{\mathrm{o}}\}$ (Eq. 7) and thus describes the image $I$ with novel objects $(C_k(I))$. Maximizing the ELBO in Eq. (6) with the data pair $\{\hat{y}, I\}$ allows to describe images with novel objects in a variational framework for diverse mappings.

### 3.3 Network architecture

We illustrate our complete network architecture in Fig. 2a. Given the image-caption paired training input, we first extract image features $\{v_1, \ldots, v_k\}$ with Faster R-CNN [35] and average the obtained features to serve as input image representation $\bar{v}$ [3, 31]. For the caption $x$, we first extract the sequence $x^{\mathrm{m}}$, replacing the occurences of objects within $x$ with the placeholder <s>. We denote the list of objects extracted from $x$ as $x^{\mathrm{o}}$. During training, we enforce our COS latent space with two LSTMs, modeling the posterior distributions $q_{\theta^{\mathrm{o}}}$ and $q_{\theta^{\mathrm{m}}}$ in sequential latent spaces, parameterized by $\theta^{\mathrm{o}}$ and $\theta^{\mathrm{m}}$ respectively. As shown in Fig. 2a, the LSTM $q_{\theta^{\mathrm{m}}}$ takes as input the sequence $x^{\mathrm{m}}$ and the image features $\bar{v}$ to encode the contextual information $z^{\mathrm{m}}$. Similarly, the LSTM $q_{\theta^{\mathrm{o}}}$ takes as input $x^{\mathrm{o}}$, $z^{\mathrm{m}}$, and image features $\bar{v}$ to encode the object information in $z^{\mathrm{o}}$. Analogously, the conditional priors $p_{\phi^{\mathrm{m}}}$ and $p_{\phi^{\mathrm{o}}}$, parameterized by $\phi^{\mathrm{m}}$ and $\phi^{\mathrm{o}}$, are modeled with LSTMs. The conditional prior $p_{\phi^{\mathrm{m}}}$ takes the image features $\bar{v}$ as input and the conditional prior $p_{\phi^{\mathrm{o}}}$ takes the image features $\bar{v}$ as well as the encoded $z^{\mathrm{m}}$ for the same caption. The priors are conditional Gaussian distributions at each timestep. We use the attention LSTM of Anderson et al. [4] as the decoder for modeling the posterior distribution. The input to the standard attention LSTM is the hidden state of the language LSTM at the previous timestep, the context vector, and the ground-truth word of the caption at timestep $t - 1$. In our model, instead of $\bar{v}$ as context vector, we input $[z_t, \bar{v}]$ to the LSTM as the context at each timestep. Thus, the feature weights processing $\{v_1, \ldots, v_k\}$ depend on the vector $[z_t, \bar{v}]$, guiding the attention LSTM to attend to certain regions in the image. The output of the attention LSTM along with the attended image features is input to the language LSTM to generate the word at timestep $t$. More details can be found in the supplemental material.

**Generative model.** As shown is Fig. 2b, given an image $I$, we first sample the latent representation $\tilde{z}^{\mathrm{m}}$ from the prior $p_{\phi^{\mathrm{m}}}$, *i.e.* $\tilde{z}_t^{\mathrm{m}} \sim \mathcal{N}(\mu_t^{\mathrm{m}}, \sigma_t^{\mathrm{m}})$ where $\mu_t^{\mathrm{m}}$, $\sigma_t^{\mathrm{m}}$ are the mean and variance of the conditional sequential prior at timestep $t$. With $\tilde{z}^{\mathrm{m}}$ and $I$, we sample $\tilde{z}^{\mathrm{o}}$ from the conditional sequential prior $p_{\phi^{\mathrm{o}}}$. $[\tilde{z}_t^{\mathrm{m}}, \tilde{z}_t^{\mathrm{o}}]$ is then input to the attention LSTM to generate a caption for the image.

## 4   Experiments

To show the advantages of our method for diverse and accurate image captioning, we perform experiments on the COCO dataset [29], consisting of $82\,783$ training and $40\,504$ validation images, each with five captions. Consistent with [6, 14, 44], we use $118\,287$ train, $4000$ validation, and $1000$ test images. We additionally perform experiments on the held-out COCO dataset [17] to show that our *COS-CVAE* framework can be extended to training on images with novel objects. This dataset is a subset of the COCO dataset and excludes all the image-text pairs containing at least one of the eight specific objects (in any one of the human annotations) in COCO: "bottle", "bus", "couch", "microwave", "pizza", "racket", "suitcase", and "zebra". The training set consists of $70\,000$ images. For this setting, COCO validation [29] is split into two equal halves for validation and test data.

Table 2: Best-1 accuracy for an oracle evaluation with different metrics on the COCO dataset.

| Method | #Samples | B-4 (↑) | B-3 (↑) | B-2 (↑) | B-1 (↑) | C (↑) | R (↑) | M (↑) | S (↑) |
|---|---|---|---|---|---|---|---|---|---|
| Div-BS [40] | | 0.383 | 0.538 | 0.687 | 0.837 | 1.405 | 0.653 | 0.357 | 0.269 |
| POS [14] | 20 | 0.449 | 0.593 | 0.737 | 0.874 | 1.468 | 0.678 | 0.365 | 0.277 |
| AG-CVAE [42] | | 0.471 | 0.573 | 0.698 | 0.834 | 1.259 | 0.638 | 0.309 | 0.244 |
| Seq-CVAE [6] | | 0.445 | 0.591 | 0.727 | 0.870 | 1.448 | 0.671 | 0.356 | 0.279 |
| Seq-CVAE (attn) | 20 | 0.486 | 0.629 | 0.762 | 0.893 | 1.599 | 0.692 | 0.261 | 0.291 |
| COS-CVAE (paired) | 20 | 0.492 | 0.634 | 0.764 | 0.895 | 1.604 | 0.700 | 0.382 | 0.291 |
| COS-CVAE | | **0.500** | **0.640** | **0.771** | **0.903** | **1.624** | **0.706** | **0.387** | **0.295** |
| Div-BS [40] | | 0.402 | 0.555 | 0.698 | 0.846 | 1.448 | 0.666 | 0.372 | 0.290 |
| POS [14] | | 0.550 | 0.672 | 0.787 | 0.909 | 1.661 | 0.725 | 0.409 | 0.311 |
| AG-CVAE [42] | 100 | 0.557 | 0.654 | 0.767 | 0.883 | 1.517 | 0.690 | 0.345 | 0.277 |
| Seq-CVAE [6] | | 0.575 | 0.691 | 0.803 | 0.922 | 1.695 | 0.733 | 0.410 | 0.320 |
| LNFMM [32] | | 0.597 | 0.695 | 0.802 | 0.920 | 1.705 | 0.729 | 0.402 | 0.316 |
| Seq-CVAE (attn) | 100 | 0.592 | 0.710 | 0.821 | 0.929 | 1.808 | 0.747 | 0.433 | 0.327 |
| COS-CVAE (paired) | 100 | 0.593 | 0.712 | 0.823 | 0.930 | 1.812 | 0.748 | 0.436 | 0.329 |
| COS-CVAE | | **0.633** | **0.739** | **0.842** | **0.942** | **1.893** | **0.770** | **0.450** | **0.339** |

**Evaluation metrics.** The accuracy of the captions is evaluated with standard metrics – Bleu (B) 1–4 [34], CIDEr [C; 38], ROUGE [R; 28], METEOR [M; 13], and SPICE [S; 1]. The diversity of the generated samples is compared using the metrics used in competing methods following the standard protocol [6, 14, 32, 44]: *Uniqueness* is the percentage of unique captions generated by sampling from the latent space. *Novel sentences* are the captions not seen in the training data. *m-Bleu-4* computes Bleu-4 for each diverse caption with respect to the remaining diverse captions per image. The Bleu-4 obtained is averaged across all images. Lower m-Bleu indicates higher diversity. *Div-n* is the ratio of distinct $n$-grams per caption to the total number of words generated per set of diverse captions.

**Evaluation on the COCO dataset.** We evaluate our approach against state-of-the-art methods for diverse image captioning. We include *DIV-BS* [40], which extends beam search and *POS* [14], with parts-of-speech supervision. We compare against variational methods: *AG-CVAE* [44], *Seq-CVAE* [6], and *LNFMM* [32]. We further include various ablations of our model to show the effectiveness of the components of our model. *Seq-CVAE (attn)* non-trivially extends *Seq-CVAE* [6], where the sampled $z_t$ at each timestep is input to the attention LSTM of Anderson et al. [1]. This baseline shows the effect of a strong decoder with the structured latent space of prior work. We further include *COS-CVAE (paired)* with only paired data for supervision (*i.e.* without pseudo supervision).

Table 2 shows the oracle performance, where the caption with maximum score per metric is chosen as best-1 [6, 44]. We consider 20 and 100 samples of $z$, consistent with prior work. The oracle evaluation scores the top sentence, but is evaluated for a limited number of sentences per image and averaged across the test set. Thus, for a high oracle score the model needs to put high probability mass on the likely captions for an image. Our complete *COS-CVAE* model, which leverages context-based pseudo supervision from the neighboring contextual descriptions, outperforms the previous state of the art by a large margin in terms of accuracy. We improve the CIDEr and Bleu score over the previous state of the art for 100 sampled captions by ∼11.0% and ∼6.0%, respectively. The average and worst CIDEr on the 20 samples for *COS-CVAE* are 0.920 and 0.354, respectively, compared to the the average and worst CIDEr of 0.875 and 0.351 for *Seq-CVAE*. The high gain in accuracy metrics shows that our *context-object split* latent space succeeds at modeling the semantics and the relevant variations in contextual descriptions for a given image in the dataset. This leads to better generalization on the test dataset. Comparing *COS-CVAE* and *COS-CVAE (paired)* further shows

Table 3: Consensus re-ranking for captioning using CIDEr on the COCO dataset.

| Method | B-4 (↑) | B-3 (↑) | B-2 (↑) | B-1 (↑) | C (↑) | R (↑) | M (↑) | S (↑) |
|---|---|---|---|---|---|---|---|---|
| Div-BS [40] | 0.325 | 0.430 | 0.569 | 0.734 | 1.034 | 0.538 | 0.255 | 0.187 |
| POS [14] | 0.316 | 0.425 | 0.569 | 0.739 | 1.045 | 0.532 | 0.255 | 0.188 |
| AG-CVAE [42] | 0.311 | 0.417 | 0.559 | 0.732 | 1.001 | 0.528 | 0.245 | 0.179 |
| LNFMM [32] | 0.318 | 0.433 | 0.582 | 0.747 | 1.055 | 0.538 | 0.247 | 0.188 |
| COS-CVAE (paired) | **0.354** | **0.473** | **0.620** | **0.777** | **1.144** | **0.563** | **0.270** | **0.204** |
| COS-CVAE | 0.348 | 0.468 | 0.616 | 0.774 | 1.120 | 0.561 | 0.267 | 0.201 |

Table 4: Qualitative comparison of captions generated by our COS-CVAE with different methods.

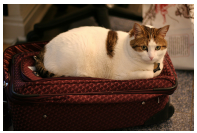

| Image | DIV-BS | Seq-CVAE | COS-CVAE (paired) | COS-CVAE |
|---|---|---|---|---|
| | • a cat is sitting in a suit-case<br>• a black and white cat sitting on top of a piece of luggage<br>• a black and white cat is sitting in a suitcase | • a cat is sitting on a suit-case on a bed<br>• cat sitting on a piece of luggage<br>• a small cat sitting on the back of a suitcase | • a close up of a cat on top of a suitcase<br>• a cat sitting on top of a piece of luggage<br>• a cat laying on top of a suitcase | • a cat sitting on top of luggage on the floor<br>• a cat curled up on a piece of luggage<br>• a close up of a very cute cat in a red suitcase |

that the annotated paired data is insufficient to fully capture the multimodality of the underlying generative process. Moreover, our context-based pseudo-supervision signal contributes to generating coherent and meaningful captions without loss of structure in the generated captions, see Tab. 4.

Table 3 shows the results on various accuracy metrics for captioning after consensus re-ranking. In consensus re-ranking, given a test image, the nearest neighboring image is found in the training set and the captions of the neighbors are used as reference to evaluate the accuracy metrics [33]. *COS-CVAE (paired)* and *COS-CVAE* outperform the previous state of the art on all accuracy metrics. The accuracy of captions with consensus re-ranking using *COS-CVAE (paired)* shows that our method can generate captions representative of the paired training data. Furthermore, the accuracy from consensus re-ranking of *COS-CVAE* is competitive to that of *COS-CVAE (paired)*, showing that even when trained with additional context-based pseudo supervision, *COS-CVAE* can model the distribution from the paired training data. However, as consensus re-ranking considers image similarities in the paired training data, generalization to unseen images, such as the test set, is not rewarded. Therefore, the benefit of context-based pseudo supervision is not fully apparent in an evaluation using consensus re-ranking. In fact, as shown in Tab. 2, *COS-CVAE* offers better generalization beyond the supervision available from the paired training data. In Fig. 3, we compare the best-10 sentences after consensus re-ranking against prior work and beam search on a standard captioning method [4], which has the same base network. *COS-CVAE* is competitive to the computationally expensive beam search and has better CIDEr scores compared to competing methods, thus is diverse *yet* accurate.

Our *COS-CVAE* model expresses high diversity in the context-object split latent space, utilizing contextual descriptions beyond that presented in the paired training data. This is reflected in the *m-BLEU-4*, *Div-1*, and *Div-2* scores (Tab. 5), where our model outperforms the previous state of the art by $\sim 11.6\%$, $\sim 5.4\%$, and $\sim 10.5\%$ respectively, showing much higher $n$-gram diversity in the generated samples of captions. Furthermore, the qualitative results in Tab. 4 show that the *COS-CVAE* generates more diverse captions, *e.g.* containing varied contextual information "curled up on a piece" or "of a very cute" compared to competing methods.

**Ablation Studies.** Next, we validate the design choice of including the attention LSTM [1]. Although *Seq-CVAE (attn)* improves the accuracy of the generated captions over *Seq-CVAE*, the diversity with respect to the uniqueness and the m-Bleu-4 decreases in comparison to *Seq-CVAE*. Our *COS-CVAE (paired)* with COS factorization and strong conditional priors allows to integrate an attention LSTM decoder in our model, and while being of comparable accuracy to *Seq-CVAE (attn)*, improves the diversity over *Seq-CVAE* and *Seq-CVAE (attn)*. This shows that our factorized

Table 5: Diversity evaluation on at most the best-5 sentences after consensus re-ranking.

| Method | Unique (↑) | Novel (↑) | mBLEU (↓) | Div-1 (↑) | Div-2 (↑) |
|---|---|---|---|---|---|
| Div-BS [40] | **100** | 3421 | 0.82 | 0.20 | 0.25 |
| POS [14] | 91.5 | 3446 | 0.67 | 0.23 | 0.33 |
| AG-CVAE [42] | 47.4 | 3069 | 0.70 | 0.23 | 0.32 |
| Seq-CVAE [6] | 84.2 | 4215 | 0.64 | 0.33 | 0.48 |
| LNFMM [32] | 97.0 | **4741** | 0.60 | 0.37 | 0.51 |
| Seq-CVAE (attn) | 61.9 | 4323 | 0.69 | 0.34 | 0.50 |
| COS-CVAE (paired) | 95.9 | 4411 | 0.67 | 0.34 | 0.50 |
| COS-CVAE | 96.3 | 4404 | **0.53** | **0.39** | **0.57** |

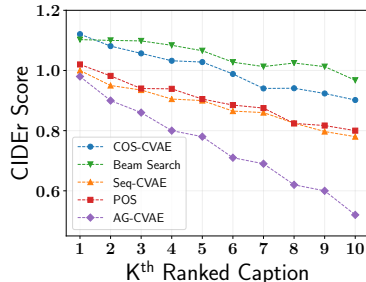

Figure 3: CIDEr score of consensus re-ranked best-10 captions from 20 samples.

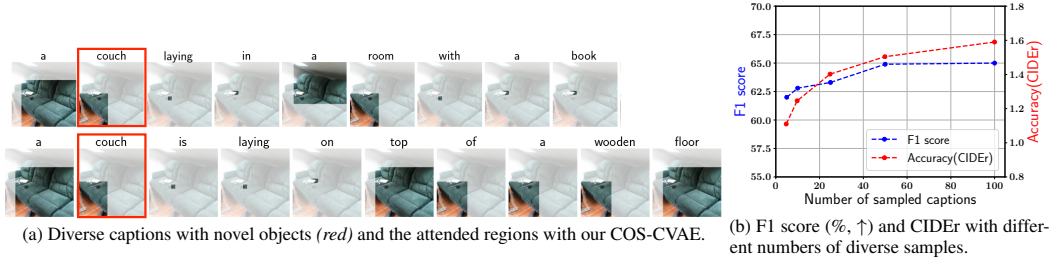

(a) Diverse captions with novel objects *(red)* and the attended regions with our COS-CVAE.

(b) F1 score (%, ↑) and CIDEr with different numbers of diverse samples.

Figure 4: COS-CVAE for captioning images with novel objects.

Table 6: F1 scores (%, ↑) for evaluation on the held-out COCO dataset.

| Method | bottle | bus | couch | microwave | pizza | racket | suitcase | zebra | average |
|---|---|---|---|---|---|---|---|---|---|
| DCC [17] | 4.6 | 29.8 | 45.9 | 28.1 | 64.6 | 52.2 | 13.2 | 79.9 | 39.8 |
| NOC [39] | 14.9 | 69.0 | 43.8 | 37.9 | 66.5 | 65.9 | 28.1 | 88.7 | 51.8 |
| NBT [31] | 7.1 | 73.7 | 34.4 | 61.9 | 59.9 | 20.2 | 42.3 | 88.5 | 48.5 |
| PS3 [3] | – | – | – | – | – | – | – | – | 63.0 |
| LSTM-P [27] | 28.7 | 75.5 | 47.1 | 51.5 | 81.9 | 47.1 | 62.6 | 93.0 | 60.9 |
| Seq-CVAE (attn, CBS) | 32.4 | 80.9 | 27.0 | 55.1 | 63.8 | 25.8 | 44.7 | 94.4 | 53.0 |
| COS-CVAE | **35.4** | **83.6** | 53.8 | **63.2** | 86.7 | **69.5** | 46.1 | 81.7 | 65.0 |
| COS-CVAE (CBS) | 33.5 | 81.9 | **58.3** | 57.8 | **88.3** | 63.4 | **65.7** | **96.2** | **68.1** |

latent space can better represent diversity even with strong posteriors. The improved accuracy of the *COS-CVAE (paired)* over *Seq-CVAE* implies that we can generate coherent captions with our COS latent space. Furthermore, the context-based pseudo supervision, which can only be leveraged with the structure of our COS factorization, where contextual variations can be modeled independently of the object information, accounts for increased diversity with a high boost in performance.

**Evaluation on novel object captioning.** Our *COS-CVAE* framework can be trained on images with novel objects (without paired training data) by exploiting contextual descriptions of captions in the dataset. At test time, we input the image and generate captions without any explicit object-based information. For reference, we include the scores from approaches for novel object captioning, which are trained to generate a single caption for a given image: *DCC* [17], *NOC* [39], *NBT* [31], *PS3* [3], and *LSTM-P* [27]. We further include *Seq-CVAE (attn, CBS)* and *COS-CVAE (CBS)* with constrained beam search using two constraints [2]. The test data statistics for held-out COCO show that ∼57% of the images with novel objects have at least one ground-truth caption without the mention of the novel object. Therefore, it would be limiting to expect every caption from a diverse captioning approach, which learns the distribution of likely captions, to mention the novel object. The mention of the novel object in any of the sampled captions for the image counts towards a positive score. From the F1 scores in Tab. 6, we observe that our *COS-CVAE* achieves competitive results with a relative improvement of ∼3.1% over methods generating a single caption for images containing novel concepts. Furthermore, the captions containing the mentions of the novel objects are coherent and exhibit diversity as shown in Fig. 4a. The high F1-score of *COS-CVAE* compared to *Seq-CVAE (attn, CBS)* highlights the benefit of training with context-based pseudo supervision for images with novel objects. Adding CBS constraints to *COS-CVAE* further improves the F1-score. We show the F1-score and the highest CIDEr for sample sizes of $z$ in $\{5, \dots, 100\}$ in Fig. 4b. Here, even when sampling 5 diverse captions, the model generates captions with mentions of novel objects with an F1 score of 62.5%, which increases by 2.5% points for 50 or 100 diverse samples. The high CIDEr score shows that the captions not only contain mentions of novel objects but are also accurate (*cf*. Fig. 4b).

## 5 Conclusion

In this paper, we presented a novel COS-CVAE framework for diverse image captioning, consisting of a context-object split latent space, which exploits diversity in contextual descriptions from the captions explaining similar contexts through pseudo supervision. This leads to increased multimodality in the latent spaces, better representing the underlying true data distribution. Furthermore, our framework is the first that allows for describing images containing novel objects in a setup for diverse image captioning. Our experiments show that COS-CVAE achieves substantial improvements over the previous state of the art, generating captions that are both diverse and accurate.

## Broader Impact

One of the promising applications of image captioning is to convert visual content on the web or from photographs to the form of text. This content can then be made more accessible to visually impaired people through text-to-speech synthesis [41]. Additionally, image captioning has been found useful in application areas such as in the medical domain, where it can provide assistance for generating diagnostics from X-ray images [37]. Diverse image captioning, as presented in our work, aims to resolve the ambiguities or uncertainties that occur while explaining visual scenes. Multiple captions help in reducing the impact of errors in explanations, since a user then has a better overview of the concepts and contexts represented in images.

Apart from such direct, positive impact on individuals or specific application domains, social media platforms utilize image captioning to mine the visual content for data summarization [24]. While this helps organizations in managing online content such as latest trends and may provide value to users of the platform, it can also compromise the users' privacy, *e.g.* summarizing user behavior or preferences for targeted advertisement. Moreover, captioning algorithms are still limited by dataset biases and the availability of exhaustive human annotations [18]. In this paper, we attempt to address the latter by leveraging annotations beyond that available from the paired training data. This is inspired from the observation that within the dataset, humans label captions of images with similar contextual information in many possible variations – by focusing on different regions of images or through different interpretations. Thus we can generate diverse captions representative of the underlying data distribution. Despite this, clearly more research is necessary to reach human-level accuracy and diversity.

## Acknowledgments and Disclosure of Funding

This work has been supported by the German Research Foundation as part of the Research Training Group *Adaptive Preparation of Information from Heterogeneous Sources (AIPHES)* under grant No. GRK 1994/1. Further, this work has in part received funding from the European Research Council (ERC) under the European Union's Horizon 2020 research and innovation programme (grant agreement No. 866008). We would like to thank the reviewers for their fruitful comments.

## Footnotes

[1]Code available at `https://github.com/visinf/cos-cvae`

[2]For simplicity of notation, written for a single data point. We assume the training dataset is *i.i.d.*

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
