[Supplementary Material]

# Diverse Image Captioning with Context-Object Split Latent Spaces
## – Supplemental Material –

**Shweta Mahajan**      **Stefan Roth**
Dept. of Computer Science, TU Darmstadt
{mahajan@aiphes, stefan.roth@visinf}.tu-darmstadt.de

## A   Implementation Details

The experiments for accuracy and diversity performed on the COCO dataset [29] are trained for 88 epochs (70 000 iterations), following previous work [3]. We use the SGD optimizer with a learning rate of 0.015, momentum 0.9, and a weight decay of 0.001. The latent spaces $z_t^{\mathrm{m}}$ and $z_t^{\mathrm{o}}$ are each of dimension 64. The image features $\bar{v}$ are obtained by averaging the ResNet-101 features of all the regions from Faster R-CNN [35], pre-trained on the Visual Genome dataset [49] using the implementation of [4]. The text encoder, parameterized by $\theta^{\mathrm{m}}$, encoding $z_t^{\mathrm{m}}$ consists of a bidirectional LSTM with a hidden size of 1024. The word dimension for the embedding layer is 300. A linear layer is applied to project the output to a 64 dimensional space. The LSTM takes as input at each timestep a concatenated vector of $\bar{v}$ and the encoded sequence in the forward direction as well as the reverse direction. Similarly, the text encoder, parameterized by $\theta^{\mathrm{o}}$, encoding $z_t^{\mathrm{o}}$ consists of an LSTM with a hidden size of 1024. It takes as input a concatenated vector of $\bar{v}$ and the contextual representation $z_t^{\mathrm{m}}$ at each timestep. The final sequential latent output is obtained by applying a fully connected layer of size 64 to the concatenation of the fixed hidden representation obtained from the LSTM of $q_\theta^{\mathrm{o}}$, the contextual representation $z_t^{\mathrm{m}}$, and the output of the previous timestep $z_{t-1}^{\mathrm{o}}$. The sequential conditional Gaussian priors are implemented with LSTM cells, parameterized by $\phi^{\mathrm{m}}$ and $\phi^{\mathrm{o}}$, with hidden size 1024 and 256 respectively. The LSTM cell encoding $p_{\phi^{\mathrm{m}}}$ takes as input the concatenation of the vectors $\bar{v}$, $x_{t-1}$, and $z_{t-1}^{\mathrm{m}}$. Similarly, the LSTM cell encoding $p_{\phi^{\mathrm{o}}}$ takes as input the concatenation of the vectors $\bar{v}$, $x_{t-1}$, $z_{t-1}^{\mathrm{m}}$, and $z_{t-1}^{\mathrm{o}}$. The attention LSTM follows the parameterization of [3]. The overall objective from Eqs. (2) and (8) of the main paper to be optimized is of the form

$$(1-\alpha)L(x,I) + \alpha\hat{L}(\hat{y},I) \quad \rightarrow \max_{\theta,\phi}.$$

Here, the value of the regularization parameter $\alpha$ that controls the objective for pseudo supervision is set to 0.2, obtained through grid search in the range $[0.1, 0.4]$ with a step size of 0.1. The size of the latent dimensions is chosen through grid search over $[64, 256]$ with a step size of 64.

## B   Additional Evaluation Details

In Tab. 7 we further evaluate the diversity of *COS-CVAE* using self-CIDEr [50], which scores caption similarity based on a latent semantic analysis. We include a comparison to beam search [4] and *Seq-CVAE (attn)*, where our approach again exhibits better diversity than the baseline methods.

For all our experiments, we performed 5 training runs with the chosen set of hyperparameters with random initializations. Furthermore, to account for the stochasticity of the oracle evaluation, we evaluated our model over 5 runs of each given trained model. While in the main paper we show the result from one run for consistency with previous work, we here additionally give the mean and standard deviation of the

Table 7: Diversity evaluation with Self-CIDEr.

| Method | Self-Cider ($\uparrow$) |
|---|---|
| Beam search | 0.58 |
| Seq-CVAE (attn) | 0.71 |
| COS-CVAE | **0.74** |

Table 8: Oracle evaluation (*cf.* Tab. 2 of the main paper), with mean and std. deviation.

| Method | B-4 (↑) | B-3 (↑) | B-2 (↑) | B-1 (↑) | C (↑) | R (↑) | M (↑) | S (↑) |
|---|---|---|---|---|---|---|---|---|
| COS-CVAE | 0.633 | 0.739 | 0.842 | 0.942 | 1.893 | 0.770 | 0.450 | 0.339 |
| COS-CVAE (CT) | 0.632±0.003 | 0.739±0.001 | 0.843±0.001 | 0.942±0.001 | 1.892±0.004 | 0.771±0.001 | 0.450±0.002 | 0.340±0.001 |

Table 9: Diversity evaluation (*cf.* Tab. 5 of the main paper), with with mean and std. deviation.

| Method | Unique (↑) | Novel (↑) | mBLEU (↓) | Div-1 (↑) | Div-2 (↑) |
|---|---|---|---|---|---|
| COS-CVAE | 96.3 | 4404 | 0.53 | 0.39 | 0.57 |
| COS-CVAE (CT) | 96.2±0.2 | 4402±30 | 0.53±0.002 | 0.39±0.00 | 0.57±0.00 |

Table 10: Evaluation for novel objects on held-out COCO (*cf.* Tab. 6 of the main paper). F1 score (%, ↑) with mean and std. deviation.

| Method | bottle | bus | couch | microwave | pizza | racket | suitcase | zebra | F1 |
|---|---|---|---|---|---|---|---|---|---|
| COS-CVAE | 35.4 | 83.6 | 53.8 | 63.2 | 86.7 | 69.5 | 46.1 | 81.7 | 65.0 |
| COS-CVAE (CT) | 35.1±0.57 | 83.1±0.30 | 53.8±0.44 | 62.9±0.83 | 86.8±0.19 | 69.0±0.62 | 47.7±0.61 | 81.3±0.49 | 64.7±0.41 |

evaluation scores of the different metrics across all runs in Tabs. 8 to 10. The top row of each table (*COS-CVAE*) shows the values from the single run that is reported in the main paper, which are within one standard deviation from the mean. The second row of each table (*COS-CVAE (CT)*) shows the central tendency (CT) with mean and standard deviation for the corresponding evaluation metrics.

## C   Additional Qualitative Examples

We provide additional qualitative results in Tabs. 11 to 13. In Tab. 11 we show the retrieved contexts with contextual descriptions representative of the visual scene and the generated pseudo captions, where the object information is obtained from the COS-CVAE framework trained on paired data.

Our COS-CVAE approach allows for diverse captioning on novel objects by modeling contextual information independent of the object information combined with the attention model of [3]. In Tab. 12 we show the divserse captions for novel objects generated by our model and the regions attended to by the attention LSTM. To the best of our knowledge, this is the first work that allows for diverse captioning on novel objects.

Table 13 provides additional qualitative examples of the diverse captions generated by the COS-CVAE approach and a comparison of the generated captions to different state-of-the-art methods [6, 32, 40].

We additionally include qualitative results for captioning on the nocaps dataset [47] in Tab. 14.[1] We trained our COS-CVAE model on the COCO dataset and included a subset of 200000 images from the Open Images [48] training set. Our COS-CVAE model generates diverse captions for images containing novel objects. Note that for a given out-of-domain image, while all the diverse captions effectively describe the visual scene, not all contain a mention of the novel object.

## D   Validation with Human Evaluation

To further evaluate the quality of the captions generated with our COS-CVAE approach, we conduct a user study following the procedure of [32]. Here, five captions for a set of images are presented to four human annotators. The annotators score the captions for each image for accuracy and diversity on the scale of 1 to 10. Here, accuracy is defined as how well the captions describe the details in a given image and diversity can be syntactic diversity or semantic diversity.

We compare our full COS-CVAE approach with pseudo supervision against the baseline COS-CVAE (paired), which is trained only on paired training data. Thus we aim to evaluate the effect of context-

Figure 5: Comparison of human accuracy and diversity scores of captions for images generated with COS-CVAE and the baseline COS-CVAE (paired): Scores for captions of an image averaged across all annotators (*left*) and scores for all captions of an image for each annotator (*right*). To make coinciding scores be easier to see, a small random jitter is added to each human assessment.

based pseudo supervision on the diverse captions. In Fig. 5 *(left)* we show the average accuracy and diversity scores again averaged across annotators; in Fig. 5 *(right)* we show the accuracy and diversity scores from each annotator. We find that the captions generated by the COS-CVAE are scored to be more accurate compared to COS-CVAE (paired). The average accuracy scores are in the range of $[7, 10]$ and $[4, 9]$ for COS-CVAE and COS-CVAE (paired), respectively. The captions generated by COS-CVAE were rated to be more diverse with scores in the range of $[7, 10]$ compared to COS-CVAE (paired), with average diversity scores in the range of $[5, 8]$.

## Footnotes

[1]The evaluation server for nocaps accepts only one caption per image and does not support methods modeling one-to-many relationships for images and captions. Working around this with consensus re-ranking [33] using the COCO training set is problematic, since the captions of out-of-domain images in the nocaps dataset containing novel objects cannot be rewarded in this way.

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

Table 11: Context-based pseudo supervision from captions of the COCO dataset.

| Image | Human annotations | Retrieved contexts | Generated pseudo captions |
|---|---|---|---|
|  | • a pizza sitting on top of a pan in an oven<br>• a pizza sitting in an oven waiting to be cooked<br>• a pizza being made inside of an oven<br>• a pizza sits on the rack of an oven uncooked<br>• an unbaked pizza sitting on a rack in the oven | • a \<s\> topped with lots of cheese and veggies<br>• a \<s\> pie with several toppings being served<br>• an uncooked \<s\> is loaded with many ingredients<br>• a \<s\> is on a lazy preparing to be served<br>• this \<s\> has cheese tomatoes bacon and peas | • a pizza topped with lots of cheese and veggies<br>• a pizza pie with several toppings being served<br>• an uncooked pizza is loaded with many ingredients<br>• a pizza is on a lazy preparing to be served<br>• this pizza has cheese tomatoes bacon and peas |
|  | • a strawberry cake with whip cream next to a some cups<br>• the desert is prepared and ready to be eaten<br>• a checkered white black bowl full of strawberry shortcake dessert<br>• a white and black plate holding a strawberry short cake<br>• a plate with a crepe covered with sliced strawberries and whipped cream | • \<s\> with fresh berries and powdered sugar on a \<s\> with a \<s\><br>• four \<s\> that are close together with writing on them<br>• a \<s\> of cereal with raspberries is sitting on a counter<br>• a beautiful platter of cheesecake and strawberries<br>• some sugary breakfast items and \<s\> on a \<s\> | • strawberries with fresh berries and powdered sugar on a plate with a cup<br>• four strawberries that are close together with writing on them<br>• a plate of cereal with raspberries is sitting on a counter<br>• a beautiful platter of cheesecake and strawberries<br>• some sugary breakfast items and strawberries on a plate |
|  | • a fire hydrant and some cars on a street<br>• fire hydrant and pylons on a city street corner<br>• a fire hydrant on city street with several parked cars<br>• there is a fire hydrant surrounded by three metal poles<br>• cars are parked on the street next to an old fire hydrant | • a coin operated \<s\> beside a trash can in a city<br>• a close up of a \<s\> on a street pole<br>• a piece of debris between a \<s\> and a trash can<br>• a grey discolored \<s\> stands on the street<br>• a \<s\> toilet sits outside on a sidewalk | • a coin operated fire hydrant beside a trash can in a city<br>• a close up of a fire hydrant on a street pole<br>• a piece of debris between a fire hydrant and a trash can<br>• a grey discolored fire hydrant stands on the street<br>• a fire hydrant toilet sits outside on a sidewalk |
|  | • a steam train is traveling down the tracks<br>• a train going along a water side area and a forested area<br>• there is a train that has passed by a river<br>• a train engine carrying carts down a mountain side by water<br>• a train driving on top of a rocky cliff | • an old \<s\> is moving along and billowing steam<br>• a \<s\> is riding across a countryside on a \<s\> track<br>• the \<s\> looks like an old fashioned one<br>• a \<s\> makes its way down a track<br>• the \<s\> is headed in the direction of the photographer | • an old train is moving along and billowing steam<br>• a train is riding across a countryside on a ground track<br>• the train looks like an old fashioned one<br>• a train makes its way down a track<br>• the train is headed in the direction of the photographer |
|  | • a group of cows grazing on grass behind a fence<br>• cows are grazing in the countryside with a bridge in the background<br>• a group of cows grazing on grass behind a fence<br>• cows are eating and grazing in the grass<br>• cows are eating and grazing in the grass | • a country side with a large grass field and \<s\> and \<s\> laying around<br>• there are many \<s\> eating at the zoo together<br>• a herd of \<s\> grazing on a lush green field<br>• black and white \<s\> graze on a grassy hill<br>• a large herd of \<s\> are grazing in a field | • a country side with a large grass field and cows laying around<br>• there are many cows eating at the zoo together<br>• a herd of cows grazing on a lush green field<br>• black and white cows graze on a grassy hill<br>• a large herd of cows are grazing in a field |

Table 12: Diverse image captioning with novel objects on the held-out COCO dataset.

Table 13: Qualitative comparison of captions generated by our COS-CVAE with different methods.

| Image | DIV-BS | Seq-CVAE | LNFMM | COS-CVAE (paired) | COS-CVAE |
|---|---|---|---|---|---|
|  | • a glass vase with some flowers in it<br>• a glass vase filled with lots of flowers<br>• a glass vase with flowers in it next to a window | • a glass vase filled with a flower in it<br>• a glass vase is sitting on a table<br>• a glass vase with flowers inside of it | • a glass of flowers on the table next to a window<br>• a vase filled with flowers in front of a window<br>• a vase with flowers sits on a table | • a glass vase on a table outside<br>• a glass of water that is sitting on a bench<br>• a glass of water is sitting on the table | • a clear glass vase is sitting on a table<br>• a glass that is sitting on a bench in the sun<br>• a glass vase is on a table near a house |
|  | • a man that is standing on a board in the water<br>• a man that is standing in the water<br>• a man that is standing on a board in a river | • a man in a water skis standing in a body of water<br>• a woman is in the water with a paddle board with a body of<br>• a man standing on a river next to a large river | • a man on a boat floating in a body of water<br>• a man riding a surfboard on a river next to trees<br>• a man on a boat floating in a body of water | • a man on a surfboard in the water<br>• a man standing on top of a surfboard in a river<br>• a man on a surfboard in the water | • a man riding a surf board down a body of water<br>• a man standing on a surf board with trees in the background<br>• a man riding on a surfboard with a paddle |
|  | • a man on a surfboard in the water with a surf board<br>• a man on a surf board in the ocean<br>• a man on a surfboard in the ocean | • a man standing next to a surfboard on the beach<br>• a man standing on a surfboard holding a surfboard<br>• a man holding a surfboard while standing on a beach | • a man in a wetsuit surfing on a surfboard<br>• a surfer is surfing on the ocean on a wave<br>• a young man rides a surfboard in a small wave | • a woman and a dog on a surfboard in the water<br>• a man holding a dog on a surfboard while standing on a beach<br>• a man holding a dog on a surfboard at the beach | • a woman and a dog on a surfboard at the beach<br>• a woman holding a dog on a surfboard in the ocean<br>• a woman on a surfboard with a black dog on a surfboard |
|  | • a group of people riding skis down a snow covered slope<br>• a group of people riding skis on top of a snow covered slope<br>• a group of people riding skis down a snow covered ski slope | • a man standing on a snowy slope while another person is riding his snowboard<br>• a group of snowboarders on a hill slope<br>• a group of people on snowboards and snowboards down | • a few people are skiing down a mountain<br>• a few people are enjoying their snowboards on a slope<br>• a few people skiing down and a snowy hill | • a group of people riding a snowboard down a snow covered slope<br>• a group of people are snowboarding down a hill<br>• a group of people are snowboarding in the snow | • a group of people riding snowboards on a snow covered slope<br>• several snowboarders are on top of a snow covered slope<br>• a group of people are in the snow with snowboards |
|  | • a close up of a small plane on a runway<br>• a black and white photo of a small plane<br>• a black and white photo of a small plane on a runway | • a black and white plane on the runway<br>• a small plane parked on a runway with<br>• a small airplane sitting on a runway next to | • a small propeller plane is on the runway<br>• a small propeller plane is sitting on a runway<br>• the airplane on the propellers is on a runway | • a small plane sitting on top of a runway<br>• a small airplane is parked on the runway<br>• an old airplane on a runway with a propeller | • a small plane sitting on the side of an airport runway<br>• an old plane sitting on the runway of an airport<br>• a small plane with a propeller tail is sitting on a runway |

Table 14: Qualitative examples of captions generated on the validation set of the nocaps [47] dataset: images with in-domain objects (*left*) and images with out-of-domain objects (*right*). The mentions of out-of-domain objects are shown in red.

| Image | Captions | Image | Captions |
|---|---|---|---|
|  | <ul><li>a truck parked next to a tree</li><li>an old black truck parked in front of a tree</li><li>an old truck that is parked in the street</li><li>a black truck parked under a tree in a park</li><li>a black truck that is parked next to a tree</li></ul> |  | <ul><li>a man and a woman standing together next to a palm tree</li><li>a woman standing by a man smiling in front of a palm tree</li><li>a couple of people posing for a photo</li><li>a couple of people that are standing near a tree</li><li>a man and woman posing for a picture</li></ul> |
|  | <ul><li>a brick building with lots of plants and a window</li><li>a brick building that has some windows outside of it</li><li>a building that is covered with lots of plants</li><li>a brick building with a red window</li><li>a brick building with a brick wall in the middle</li></ul> |  | <ul><li>a kitchen with stainless steel appliances and a metal pot</li><li>a large kitchen with a stainless steel pot</li><li>there is a large metal pot in a kitchen</li><li>a kitchen with stainless steel appliances in the middle of a kitchen</li><li>a kitchen in the middle of a restaurant with a stainless steel machine</li></ul> |
|  | <ul><li>a cat is laying in front of a parked car</li><li>a cat sleeping in front of a parked parked in a parking lot</li><li>a cat laying on the side of a car next to a car</li><li>a cat laying in front of a parked car</li><li>a cat laying on the ground in front of a black car</li></ul> |  | <ul><li>a hotel room with a bed and two lamps</li><li>a hotel room that has a bed and two lamps</li><li>there is a bed in a room with two lamps</li><li>a bedroom that has a bed and two lamps on top of it</li><li>there is a bed in a room with two lamps</li></ul> |
|  | <ul><li>a couple of trash cans sitting next to each other</li><li>a trash can sitting next to a blue trash</li><li>a trash can next to a trash</li><li>a black trash can sitting next to a trash can</li><li>a garbage can is sitting next to a trash can</li></ul> |  | <ul><li>a beer sitting on a table with a glass</li><li>a glass of beer is on a table next to a beer</li><li>a glass of a beer is sitting on a table</li><li>a beer sitting on a wooden table with a glass</li><li>a close up of a beer</li></ul> |
|  | <ul><li>a red tower with a clock on top of it</li><li>a clock tower in the middle of a red tower</li><li>a red tower with a clock in the middle of it</li><li>a clock is standing in the middle of a red building</li><li>a red tower with a red clock tower</li></ul> |  | <ul><li>a bunch of desserts are sitting on a table</li><li>a bunch of desserts sitting on a plate with some bananas</li><li>a plate filled with bananas and some food</li><li>there are some different types of food on the plate</li><li>a bunch of plates of food sitting on a white plate</li></ul> |