[Reviews · NeurIPS 2020]

Review 1

Summary and Contributions: This paper presents an approach for diverse image captioning using context-object split latent space. The authors claim that the use of context provides pseudo-supervision which promotes diversity in the generated captions. Additionally, the authors show that their approach works well for captioning novel objects that were not seen in the training data. The results provided in terms of accuracy and diversity metrics support the claims.

Strengths: The paper tackles a pertinent problem of diverse image captioning with a clear presentation of the approach and the experiments.

Weaknesses: 1.The approach is heavily inspired by the recent work on diverse image captioing using Sequential Latent spaces by Aneja et al, 2019. Although the idea of splitting object and rest of the caption is interesting but not novel in my opinion. 2. One major shortcoming is the absence of results on the recent nocaps dataset by Agrawal et al, 2019. Providing this is a crucial component of a paper that claims to be doing novel image captioining.

Correctness: yes

Clarity: yes

Relation to Prior Work: yes

Reproducibility: Yes

Additional Feedback: --post rebuttal: I feel the authours have addressed the concerns about novelty. Absence of nocaps numbers could have been addressed better by choosing 1 caption using consensus re-ranking. Having said that, the paper indeed presents exhaustive experiments and most of them improve results over baselines. I am willing to upgrade my rating.


Review 2

Summary and Contributions: =======UPDATE======= After reading the other reviews and the rebuttal, I am in favor of accepting this paper to NeurIPS. A few small notes; in the author feedback, authors note that: "The oracle evaluation has been used extensively in prior work on multi-prediction" in response to a question about if this metric is a fair comparison. Though I understand this, I think it is important that we don't get stuck on bad metrics just because it was done in prior work. I really appreciate the authors expanded thoughts on the metric in the rebuttal, and am happy they will move the consensus re-ranking (which seems like a better metric to me) to the main paper. I also appreciate the notes on why the comparison is fair for the novel object captioning and would encourage authors to include this in a final version. =======INITIAL REVIEW==== This paper proposes the COS-CVAE model which introduces a latent variable to capture diversity in the way people captions images. Unlike prior work, the latent space is split into "context" and "object" variables so additional "context" from text annotations of other images can be used as supervision. The COS-CVAE model leads to more diverse captions, and it is also able to caption novel objects. The method and application are both interesting, though I have some concerns about evaluation.

Strengths: 1. The COS-CVAE model tackles an important problem in image captioning: diversity of captions. It additionally considers another problem in image captioning, novel object captioning. I usually see papers that focus on one challenge or the other, so I really like that this paper tackles two issues in current captioning systems. The paper is relevant to the NeurIPS community. 2. Overall the method is explained clearly. The motivation for the task and model design are easy to understand. 3. There are many evaluations that allow readers to understand how the model works, including standard evaluations on diversity, sentence quality, and F1 score (for novel object captioning). 4. Some ablations are included; e.g., ablation in Tab 3 where training is restricted only to pairs of captions demonstrates why the context captions are important.

Weaknesses: My main issues are with some of the evaluations in the paper: 1. Oracle accuracy is a bit of a cheat as it scores all proposed sentences and selects the top scoring sentence. I notice that consensus re-ranking is also reported in the supplemental. The results on this are good in comparison to prior work, so I am not sure why this is not mentioned in the paper (or if the results could be squeezed into the paper by rearranging Table 3). However, even the consensus based ranking is a bit odd since it relies on finding nearest neighbor train images (how are nearest neighbors found? Stronger networks will do a better job won't they?). Additionally, there is no real comparison to "standard" caption systems (e.g., attention on attention: https://arxiv.org/pdf/1908.06954.pdf). I would like to see an evaluation where the sentences from the COS-CVAE are ranked in a similar way as is done in standard beam search (e.g., ranked by likelihood from a decoder) and compared to standard caption models. This would be helpful to understand how much we "lose" (or if we lose at all) in terms of sentence quality when compared to standard captioning systems that do not aim to generate diverse captions. From my understanding this kind of evaluation (though present in some earlier works studying diverse captions) is not present in other recent work like LNFMM. I think this is bad precedent. 2. Evaluation on novel object captioning: In L305, it is mentioned that if *any* caption mentions the novel object, this is counted as a mention. This seems extremely unfair. It would be better if one caption were chosen (e.g., ranked by the likelihood from the decoder). Evaluation would be done in the same way as is done in prior work. Additionally, no comparison to prior work is done for sentence quality. The CIDEr is much higher in recent work [26]. Furthermore, table 3b does not make much sense -- is the cider again from the highest scoring sentence? does this sentence mention the novel object? Since this is the first paper I know of that looks at both diverse captioning and novel object captioning, it is important to get the evaluation right as others will follow the standard set in this paper. 3. The paper splits sentence into context and objects; verbs, adjectives, and spatial relationships are also important for captioning models! It would be helpful if the authors could address whether splitting the annotations in such a way might lead to less accurate understanding of these types of words. The standard caption evaluations may not pick up on these smaller (but important) aspects of captions. This is especially important because of the way the model is evaluated; it is possible that one output sentence (of the 100 output) includes the correct verbs/adjectives/relations by chance without "understanding" these concepts. Since the main evaluation for accuracy is the "oracle" metric, this will not be reflected in lower cider/bleu scores.

Correctness: Overall, the claims seem to be correct though I have issues with evaluation outlined above.

Clarity: I am not clear on how the comparison to prior work in Tab 3 is done. In L241, it is mentioned that the split used to train the model follows [6, 14, 42] with 4000 validation images and 1000 test images. However, my understanding is that some of the prior work compared to in table 3 uses a different split (e.g., Div Beam Search [39] mentions the "Karpathy" split [22]). Where do the numbers for prior work in Table 3 come from? They don't seem to come from original papers. Are we sure that things like test splits are consistent across different models? Additionally, methods like Div-BS are purely decoding methods (no change to model architecture), so things like improved base networks should increase performance. Is the base vision network similar across prior work compared in Tab 3?

Relation to Prior Work: For novel object captioning, could also consider the nocaps dataset: https://nocaps.org/

Reproducibility: Yes

Additional Feedback:


Review 3

Summary and Contributions: This paper proposes a conditional variational autoencoder model to generate diverse image captions given one image, where a generated caption is controlled by the detected objects and a contextual description. The proposed model can be extended to novel object image captioning. In terms of the experiments, the proposed model achieves state-of-the-art performance on oracle test and the diversity of the generated captions still outperforms existing models.

Strengths: S1: the proposed model splits human annotations into objects that occur in images and contextual descriptions, which is novel to me. S2: the proposed model achieves state-of-the-art performance on oracle evaluation. S3: the diversity of the generated captions is relatively high compared to its counterparts.

Weaknesses: W1: the proposed model is similar to seq-VAE in the VAE part, so it would be better to demonstrate the difference between the proposed model and seq-VAE. W2: it would be better to show self-CIDEr diversity score [1]. [1] Q. Wang and A. B. Chan. Describing like humans: on diversity in image captioning. CVPR, 2019. W3: in tabel 3, best-1 accuracy is shown, which is not enough, it would be better to show both best-1 accuracy and the average metric scores, so we can see how well the proosed model performs, since sometimes too much diversity could hurt accuracy. W4: missing reference. [2] Q. Wang and A. B. Chan. Towards Diverse and Accurate Image Captions via Reinforcing Determinantal Point Process. ArXiv, 2019. W5: it would better to conduct human evaluation on the generated captions. W6: it seems that using peudo supervision could significantly improve the performance, however, peudo supervision is not the ground-truth, why it benefit the performance? How about only using psedo supervion? W7: peuso supervion is from retrieval, does it improve retrieval performance? For example, the authors could show R@K score based on the trained VSE++ model, where the query a generated caption and the condidates are images.

Correctness: Yes.

Clarity: yes.

Relation to Prior Work: see the weakness.

Reproducibility: Yes

Additional Feedback:


Review 4

Summary and Contributions: The paper addresses the problem of diverse image captioning by extending the previous VAE based approaches by factorizing the sequential latent space into two parts - one half encodes the object description at each time step while the other half captures the "context" (background, spatial relations, attributes etc). The proposed framework shows gains in diversity on the COCO dataset and also shows extension to novel object captioning.

Strengths: - The factorization of the sequential latent space to explicitly encode objects present in the image and other information such as background and spatial relationship between objects is interesting. Similar to approaches like Neural Baby Talk, it allows breaking down the image captioning approach into figuring out the general template for the sentence and the objects that will be talked about separately. - Another advantage of this approach is that it can then be extended to the problem of novel object captioning where the latent space responsible for guiding the model to attend over regions that contain novel objects. - Lastly, the context-object split of the latent space allows the model to leverage captions from similar images. Because training the model end-to-end will need "target-captions", I liked how the authors were able to use the conditional independence assumptions between the two priors to identify target objects to fill the slots in the neighboring contextual descriptions! The results show that using contextual descriptions give gains over just simply using the split latent space.

Weaknesses: 1. Avg / Worst scores: While the model outperform existing methods by producing very high CIDEr scores, I wonder if this is partly due to just producing more diverse captions with high variance in quality of each caption. Showing average / worst CIDEr / SPICE scores will convince the readers that the captions are diverse yet accurate. 2. The best-1 accuracy for oracle evaluation is considered with 100 samples, while previous works have shown these results on both 20 samples as well as 100 samples. Similarly, the diversity numbers are shown for best-5 sentences after consensus re-ranking while previous papers have shown results on both consensus re-ranking as well as all 100 samples. 3. Regarding evaluation on the novel object captioning dataset, it seems the results in the table are not fully comparable. While all the methods in Table 5 generated a single caption, COS-CVAE show average results from multiple captions. Additionally, an interesting comparison would have been to use something like constrained beam search after producing diverse captions from a simple model like "Seq-CVAE [attn]" 4. Even though it makes sense to compare their method with existing literature on the held-out COCO dataset, I would have also really liked to see numbers on the more recent nocaps* benchmark. The nocaps benchmark is much more challenging than the toy dataset and would make the arguments about novel object captioning more convincing.

Correctness: The paper shows results on standard benchmarks (COCO for Diverse Image Captioning) and the held-out COCO dataset introduced in earlier works for novel object captioning. While I found the experimentation setup to be exhaustive, there are a couple of clarifications that need to be made. See point 1, 2, 3 and 4 in weakness.

Clarity: I found the paper to be clearly written.

Relation to Prior Work: The paper borrow and extend ideas from several papers but fails to make those connections at appropriate places. For instance, the use of sequential latent spaces is borrowed from Seq-CVAE[6], the use of replacing words in the descriptions with placeholders has connections with NBT[30] and using the modified captions by replacing placeholders with novel objects for training has connections with PS3[3]. I'd recommend that instead of just clubbing all these works in related work, explicitly talk about similarities and differences while explaining relevant concepts in the approach section. *nocaps: novel object captioning at scale; ICCV 2019

Reproducibility: Yes

Additional Feedback: Update After rebuttal: Thanks for the addressing the concerns in the rebuttal. Even though I agree that since nocaps is behind an evaluation server, diversity analysis on nocaps isn't possible but I still encourage the authors to submit their top-1 caption (through consensus re-ranking) to the evaluation server. One of the goals of generating many diverse captions is to also generate a better top-1 caption so it would still be useful to take the top-1 caption and submit it to the nocaps leaderboard. My concerns around not reporting avg/worst cider were addressed in the rebuttal and should be put into the main paper. I am also satisfied with the clarifications around diversity metrics. I am recommending an accept for the paper.

[Author Response · NeurIPS 2020]

We thank all the reviewers for their helpful comments and for recognizing the novelty of our approach (R2–4) and its
potential for positive impact (R2, R4). We are glad that the reviewers found our experimental setup exhaustive (R1–4).

**Technical contribution (R1, R3).** The novelty of our approach lies in the manner in which we formulate our COS latent
space in Eq. (3) to encode the object and context information, allowing to better capture the underlying distribution.
This enables the exploitation of contextual similarities between images and captions beyond that from paired training
data in our pseudo-supervised setup. This is not feasible with prior work, *e.g*. Seq-CVAE [6]. This is the first work that
addresses diversity *and* novel object captioning in a single framework (Sec. 3.2, Eq. (7), ll. 198–212).

**Oracle evaluation and consensus re-ranking (R2).** The oracle evaluation has been used extensively in prior work
on multi-prediction (Guzmán-Rivera et al., NIPS 2012). The oracle evaluation does score the top sentence, but is
evaluated for a limited number of sentences per image and averaged across the test set. Thus, for a high oracle score the
model needs to put high probability mass on the likely captions for an image without relying on "chance". The oracle
score provides an upper bound to any re-ranking method. While consensus re-ranking can indeed be improved using
stronger networks to close the gap between oracle performance and consensus re-ranking, we follow the same consensus
re-ranking procedure as [6,14] for fairness and achieve much better accuracy (Tab. 9, suppl.). Furthermore, the accuracy
after re-ranking (Fig. 4) is competitive to methods that generate a single caption, *e.g*. [4], using beam search. We will
move Tab. 9 (suppl.) to the main paper. We compute $p(x|z, I)$ and, therefore, ranking captions $x$ for an image $I$ by
decoder probabilities $p(x|I)$ as in beam search would require marginalization over $z$, which is computationally difficult.

**Accuracy scores (R2, R3, R4).** Consensus re-ranking provides a realistic evaluation
against the competing methods (Tab. 9). To further validate our approach, we compare the
best-10 sentences after consensus re-ranking in Fig. 4 against Seq-CVAE [6] and beam
search on a standard captioning method of [4], which has the same base network. The
accuracy results show that COS-CVAE is competitive to beam search and thus diverse
*yet* accurate. The average(worst) CIDEr for all the 20 samples with COS-CVAE is
0.875(0.457) compared to 0.792(0.351) for Seq-CVAE. We will add this.

Figure 4: CIDEr score of consensus re-ranked best-10 captions from 20 samples.

**F1-score, Fig. 3b (R2, R4).** Ranking captions by decoder probabilities is not feasible
as discussed above. Consensus re-ranking is not possible as the training data does not
contain novel objects. The test data statistics for held-out COCO [17] show that $\sim$57%
of the images with novel objects have at least one ground-truth caption *without* the mention of novel object. For
ground-truth annotations, mention of novel object in any of the sentences is counted towards the F1-score [3]. Therefore,
it would be limiting to expect every caption from a diverse captioning approach, which learns the distribution of the
likely captions, to mention the novel object (Tab. 11). We will clarify and highlight these challenges in the final version.
We thus show the F1-score and the highest CIDEr for sample sizes of $z$ in $\{5, \ldots, 100\}$ in Fig. 3b and demonstrate
competitive results even for a small sample size of 5. Our COS-CVAE with 5 samples has the highest CIDEr of 1.111
and an average CIDEr of 0.775, which is competitive to NBT [30] with a similar base network. We will add this.

**Effect of context-object split (R2).** Verbs, adjectives, and spatial relationships are crucial components of the context
(ll. 105). Random samples in Tab. 2, 11, and 12 show that the captions from COS-CVAE are coherent (ll. 287–288).

**Self-CIDEr (R3).** COS-CVAE has a score of 0.742 while Seq-CVAE(attn) has 0.714. We will add a human evaluation.

**Diversity metrics, Tab. 4 (R4).** Consistent with prior work, *e.g*. [6], Unique and Novel are computed with 100 samples
and Div-1, Div-2, and m-Bleu are computed with top-5 generated captions after consensus re-ranking (ll. 251–253).

**Benefit of pseudo supervision (R3).** Pseudo supervision allows to leverage additional contextual descriptions from
human annotations that can describe the image, thus capturing the true diversity of the underlying distribution (ll. 94–96).
We use pseudo supervision along with ground-truth captions for best performance because ground-truth captions help
accurately guide the attention of the decoder (ll. 128–130). The application of pseudo supervision to a retrieval task is
an interesting idea and can be considered for future work since the extension is non-trivial.

**CBS constraints (R4).** Seq-CVAE (attn)+CBS yields an F1-score of 55.4 with 5 samples compared to an F1-score of
70.1 with COS-CVAE+CBS. However, CBS decoding leads to less diverse captions (one of the main goals here).

**Clarifications (R2).** The results of Div-BS are on the same test split as [6, 14] (*cf*. Tab. 1 in [6]). The base vision
network for different methods can be different as various models impose different constraints in the latent space [14,
41]. We have included the Seq-CVAE(attn) baseline with the same base vision network as our COS-CVAE.

**Nocaps dataset (R1, R2, R4).** Our variational COS-CVAE framework models the distribution of likely captions for a
given image. The evaluation server for the nocaps dataset (Agrawal et al.), however, accepts only one caption per image
and does not support methods modeling one-to-many relationships for images and captions. We thus provide the results
on the real world held-out COCO dataset for novel object captioning [17] instead.

**Additional references (R2, R3).** Thank you. We will add this.

[Meta-Review · NeurIPS 2020]

All reviewers recommend accept (one indicated it only in the discussion but did not update their score). The reviewers appreciate the author response and value the paper for its contributions including - the problem addressed - the idea and method to split context and objects - the extensive evaluation I agree with this evaluation and accept, however, I expect the authors to include the clarifications and improvements suggested by the reviewers and made in the author response. I also encourage the authors to include the results on nocaps as suggested by R4.